# COVID-19 Outcomes in Patients with Hematologic Malignancies in the Era of COVID-19 Vaccination and the Omicron Variant

**DOI:** 10.3390/cancers16020379

**Published:** 2024-01-16

**Authors:** Joaquín Martínez-López, Javier de la Cruz, Rodrigo Gil-Manso, Víctor Jiménez Yuste, José María Aspa-Cilleruelo, Cristian Escolano Escobar, Javier López-Jiménez, Rafael Duarte, Cristina Jacome Yerovi, José-Ángel Hernández-Rivas, Regina Herráez, Keina Quiroz-Cervantes, Rosalía Bustelos-Rodriguez, Celina Benavente, Pilar Martínez Barranco, Mariana Bastos Oteiro, Adrián Alegre, Jaime Pérez-Oteyza, Elena Ruiz, Eriel Alexis Marcheco-Pupo, Ángel Cedillo, Teresa de Soto Álvarez, Patricia García Ramirez, Rosalía Alonso Trillo, Pilar Herrera, María Luisa Bengochea Casado, Andrés Arroyo Barea, Jose Manuel Martin De Bustamante, Javier Ortiz, María Calbacho Robles, Julio García-Suárez

**Affiliations:** 1Hematology Department, Hospital Universitario 12 de Octubre, imas12 Madrid, Universidad Complutense, CNIO-ISCIII, CIBERONC, 28041 Madrid, Spain; rodrigo.gil@salud.madrid.org (R.G.-M.); ndarroy@ucm.es (A.A.B.); maria.calbacho@salud.madrid.org (M.C.R.); 2imas12 Research Institute, Hospital Universitario 12 de Octubre, 28041 Madrid, Spain; javier.delacruz@salud.madrid.org; 3Hematology Department, Hospital Universitario La Paz, 28046 Madrid, Spain; vjimenezy@salud.madrid.org (V.J.Y.); teresa.desoto@salud.madrid.org (T.d.S.Á.); josemanuel.martin@colaboradorhla.com (J.M.M.D.B.); 4Hospital Universitario Príncipe de Asturias, Universidad de Alcalá, 28805 Madrid, Spain; jose.cilleruelo@nhs.net (J.M.A.-C.); pgarciaramirez@salud.madrid.org (P.G.R.); jgarciasu.hupa@salud.madrid.org (J.G.-S.); 5Hematology Department, Hospital Universitario de Getafe, 28905 Madrid, Spain; cristian.escolano@salud.madrid.org (C.E.E.); rosalia.alonso@salud.madrid.org (R.A.T.); 6Hematology Department, Hospital Universitario Ramón Y Cajal, 28034 Madrid, Spain; jljimenez@salud.madrid.org (J.L.-J.); pilar.herrera@salud.madrid.org (P.H.); 7Hematology Department, Hospital Universitario Puerta de Hierro Majadahonda, 28222 Madrid, Spain; rafael.duarte@salud.madrid.org; 8Hematology Department, Hospital Universitario Severo Ochoa, 28914 Madrid, Spain; cristina.jacome@salud.madrid.org (C.J.Y.); marialuisa.bengochea@salud.madrid.org (M.L.B.C.); 9Hematology Department, Hospital Universitario Infanta Leonor, 28031 Madrid, Spain; jahernandezr@salud.madrid.org; 10Hematology Department, Hospital Universitario Infanta Sofía, 28702 Madrid, Spain; mariaregina.herraez@salud.madrid.org; 11Hematology Department, Hospital Universitario de Móstoles, 28935 Madrid, Spain; keinasusana.quiroz@salud.madrid.org; 12Hematology Department, Hospital Universitario del Sureste, 28500 Madrid, Spain; rosalia.bustelos@salud.madrid.org; 13Hematology Department, Hospital Clínico San Carlos, 28040 Madrid, Spain; celinamaria.benavente@salud.madrid.org; 14Hematology Department, Hospital Universitario Fundación Alcorcón, 28922 Madrid, Spain; pmartinezbarranco@salud.madrid.org; 15Hematology Department, Hospital General Universitario Gregorio Marañón, Instituto de Investigación Sanitaria Gregorio Marañón, 28009 Madrid, Spain; marianabeatriz.bastos@salud.madrid.org; 16Hematology Department, Hospital Universitario de La Princesa, IIS-HUP, 28006 Madrid, Spain; adrian.alegre@salud.madrid.org (A.A.); jortiz@salud.madrid.org (J.O.); 17Hematology Department, Hospital Universitario HM Sanchinarro, 28050 Madrid, Spain; jperezoteyza@hmhospitales.com; 18Hematology Department, Hospital Universitario del Tajo, 28300 Madrid, Spain; elena.ruizsainz@salud.madrid.org; 19Hematology Department, Hospital Universitario Infanta Cristina, 28981 Madrid, Spain; erielalexis.marcheco@salud.madrid.org; 20Asociación Madrileña de Hematología y Hemoterapia (AMHH), 28010 Madrid, Spain; st@hematologiamadrid.org

**Keywords:** COVID-19, hematologic malignancy, multicenter study, omicron variant, multiple myeloma, acute leukemia, lymphoma

## Abstract

**Simple Summary:**

This HEMATO-MADRID COVID-19 study assessed COVID-19 outcomes in 1818 hematologic cancer patients from February 2020 to October 2022 across different phases, including the Omicron period. Severe cases were more common in patients over 70 years with comorbidities or chronic lymphocytic leukemia. However, during the Omicron period, rates of severe illness reduced notably, especially among vaccinated individuals. Hospitalization, intensive care admissions, and overall mortality decreased in the Omicron phase compared to pre-Omicron, yet mortality rates in hospitalized patients remained high. Older age consistently correlated with higher mortality risk in both phases. Factors like prior stem cell transplantation, vaccination, and specific treatments were linked to improved survival rates among hematologic cancer patients facing COVID-19.

**Abstract:**

A greater understanding of clinical trends in COVID-19 outcomes among patients with hematologic malignancies (HM) over the course of the pandemic, particularly the Omicron era, is needed. This ongoing, observational, and registry-based study with prospective data collection evaluated COVID-19 clinical severity and mortality in 1818 adult HM patients diagnosed with COVID-19 between 27 February 2020 and 1 October 2022, at 31 centers in the Madrid region of Spain. Of these, 1281 (70.5%) and 537 (29.5%) were reported in the pre-Omicron and Omicron periods, respectively. Overall, patients aged ≥70 years (odds ratio 2.16, 95% CI 1.64–2.87), with >1 comorbidity (2.44, 1.85–3.21), or with an underlying HM of chronic lymphocytic leukemia (1.64, 1.19–2.27), had greater odds of severe/critical COVID-19; odds were lower during the Omicron BA.1/BA.2 (0.28, 0.2–0.37) or BA.4/BA.5 (0.13, 0.08–0.19) periods and among patients vaccinated with one or two (0.51, 0.34–0.75) or three or four (0.22, 0.16–0.29) doses. The hospitalization rate (75.3% [963/1279], 35.7% [191/535]), rate of intensive care admission (30.0% [289/963], 14.7% [28/191]), and mortality rate overall (31.9% [409/1281], 9.9% [53/536]) and in hospitalized patients (41.3% [398/963], 22.0% [42/191]) decreased from the pre-Omicron to Omicron period. Age ≥70 years was the only factor associated with higher mortality risk in both the pre-Omicron (hazard ratio 2.57, 95% CI 2.03–3.25) and Omicron (3.19, 95% CI 1.59–6.42) periods. Receipt of prior stem cell transplantation, COVID-19 vaccination(s), and treatment with nirmatrelvir/ritonavir or remdesivir were associated with greater survival rates. In conclusion, COVID-19 mortality in HM patients has decreased considerably in the Omicron period; however, mortality in hospitalized HM patients remains high. Specific studies should be undertaken to test new treatments and preventive interventions in HM patients.

## 1. Introduction

The COVID-19 pandemic caused by the SARS-CoV-2 virus has resulted in more than 750 million cases of COVID-19 worldwide, including more than 6.9 million deaths [1]. Adult patients with hematologic malignancies (HM) have been more substantially affected by COVID-19 than the general adult population, and have experienced a higher mortality rate, as they have greater susceptibility to SARS-CoV-2 infection and severe disease because of their immune-deficient status and their use of immunosuppressive treatments [2].

Multiple reports have been published on COVID-19 outcomes in patients with HM [3,4,5,6,7,8,9,10], but most of these have been focused on cases occurring during the pre-vaccination era, between March and December 2020. Therefore, changes in outcomes over time during the vaccination era and the multiple waves driven by SARS-CoV-2 variants and subvariants are not fully understood. Epidemiologic data from the pre-vaccination era showed that the mortality rate among HM patients was in the range of 13–37% overall and 19–46% for those hospitalized with COVID-19, with high rates reported particularly for patients aged ≥60 years, those with a diagnosis of acute myeloid leukemia (AML) or myelodysplastic syndrome (MDS), and patients receiving active treatment with conventional chemotherapy or monoclonal antibodies [11]. 

Data from a number of reports in the general population have demonstrated that COVID-19 mortality rates have declined over time, likely reflecting advances in detection, the development of effective COVID-19-directed therapies such as anti-SARS-CoV-2 monoclonal antibodies and new antiviral agents [12,13,14,15,16,17,18], the emergence of less virulent SARS-CoV-2 variants, and the introduction of messenger RNA (mRNA) COVID-19 vaccines (BNT162b2 and mRNA-1273) since December 2020 [19,20,21,22]. However, there have been few direct comparisons of COVID-19 severity and outcomes in HM patients between the pre-vaccination and vaccination eras. Furthermore, the impact of the highly transmissible Omicron variant on HM patients, and the protection conferred by vaccination and boosters against the severe clinical outcomes associated with Omicron, remain unknown. 

Therefore, in order to better understand clinical trends in COVID-19 outcomes among HM patients, it is imperative to consider data from across the multiple waves driven by different dominant circulating variants, and particularly from those driven by the most recent Omicron subvariants [23,24]. To date, three real-world studies have been published on COVID-19 among HM patients, including two conducted during Alpha- and Delta-dominant periods [25,26] and a third conducted during the Omicron-dominant period [27]. These studies showed high rates of hospitalization (37–53%) and death (5.7–9.2%) due to COVID-19 among HM patients. The last study focused on the Omicron-dominant period and reported an overall 16.5% mortality in hospitalized patients; older age and active malignancy increased mortality; and three doses of the vaccine were protective against progression to critical disease [27]. Although the comparison of mortality rates across studies is not straightforward due the heterogeneity in methods, the most recent studies reported rates that are lower than those during the pre-vaccination era [9,10] but that are nevertheless higher than those previously reported in the fully vaccinated general population [28].

In this context, with a rapidly evolving landscape and the initial evidence regarding the impact of COVID-19 on HM patients having become outdated, we evaluated morbidity and mortality over time from March 2020 to September 2022 in HM patients diagnosed with COVID-19 in the Madrid region of Spain, comparing outcomes between the pre-Omicron and Omicron eras and between the pre-vaccination and vaccination eras. We also analyzed the differing patient characteristics and risk factors associated with severe outcome and death, and the impact of COVID-19 therapies on outcomes.

## 2. Methods

### 2.1. Study Design and Participants

HEMATO-MADRID COVID-19 is an ongoing, observational, multicenter, and registry-based study with prospective data collection that is sponsored by the Madrid Society of Hematology (Asociación Madrileña de Hematología y Hemoterapia, AMHH) [11]. Full methodological details for this study have been reported previously [5]. Briefly, the study population was accrued from 32 healthcare centers with AMHH-affiliated hematologists in the Madrid region in Spain, covering 6.6 million inhabitants. For inclusion in the analysis, HM patients had to be aged ≥18 years and to have had a SARS-CoV-2 infection confirmed by reverse transcription–polymerase chain reaction of a nasopharyngeal swab [29] in the emergency departments, hospital wards (infection while hospitalized), or outpatient clinics of participating healthcare centers. Patients also required a medical history of HM at any time; their disease could be either active or in remission at the time of COVID-19 diagnosis, which was established based on World Health Organization (WHO) recommendations [30]. Investigators at each participating institution evaluated patients per local practice, when clinically indicated.

This study was granted by the Fundación Madrileña de Hematología y Hemoterapia and the Fundación Leucemia y Linfoma. The study protocol was approved by the local Ethics Committee and written informed consent was waived (CEIm Hospital 12 de Octubre, Spain: ref. 20/182; date of approval: 20 April 2020).

### 2.2. Study Outcomes and Data Collection

COVID-19 clinical severity and mortality, including overall survival and 30-day and 60-day survival probability estimates, were the key study endpoints. Disease severity was assessed within 24 h of admission per World Health Organization guidelines [30], with hospital/intensive care unit (ICU) admissions determined locally based on criteria updated daily during the healthcare emergency period. Data were collected through to the time of last follow-up visit or death. The key determinants evaluated for their impact on COVID-19 outcomes included pre-infection patient characteristics, type of HM and treatment received, and aspects of COVID-19 management.

The HEMATO-MADRID COVID-19 registry incorporates deidentified data on factors of relevance to patients with HM and COVID-19. For this analysis, we extracted data on age, sex, and number of specific comorbidities associated with COVID-19; these were cardiac disease, pulmonary disease (not including lung cancer), renal disease, diabetes, hypertension, and body mass index ≥ 35 kg/m^2^. We also collected data on the type of HM and therapy received. For this analysis, patients were defined as having ‘active antineoplastic treatment’ if they had received anticancer therapy within 30 days prior to their COVID-19 diagnosis. These therapies were classified as ‘conventional chemotherapy’, ‘low-intensity chemotherapy’, ‘molecular-targeted therapy’, ‘immunotherapy’, ‘immunomodulator drugs’, ‘hypomethylating agents’, or ‘supportive therapy’. Information on COVID-19 management was also extracted.

The analysis time period was sub-divided into four periods, defined according to the dominant circulating SARS-CoV-2 variant (>50% of national circulating SARS-CoV-2 lineages among recorded infections). The first period covered the wave in which the D614G SARS-CoV-2 variant was dominant and included HM patients diagnosed with COVID-19 between 27 February 2020 and 15 February 2021. The second period covered the Alpha- and Delta-dominant waves and included patients diagnosed between 15 February and 15 December 2021. The third and fourth periods covered the Omicron BA.1/BA.2-dominant and Omicron BA.4/BA.5-dominant waves, which included patients diagnosed between 15 December 2021 and 31 May 2022, and between 1 June and 30 September 2022 [31,32]. Analyses were also conducted pooling the pre-Omicron (first and second periods) and Omicron (third and fourth periods) time periods.

Eligible patients who were entered into the AMHH registry by local investigators between 27 February 2020 and 1 October 2022 were included in the analysis, with all records updated through to the end of September 2022. Patients could be added to the database at any time during their COVID-19 disease course. The study steering committee, with expertise in the research topic and in the study of HM and infectious diseases, reviewed each registered case for completeness and consistency.

### 2.3. Statistical Analysis

Patient- and disease-related factors were characterized overall, by COVID-19 severity (mild/moderate and severe/critical disease, which covered from severe pneumonia to septic shock), and by time of diagnosis (pre-Omicron and Omicron periods). Absolute and relative frequencies were calculated for all determinant factors, as well as the median and interquartile range (IQR) of patients’ ages, for all groups analyzed. Available sample size was reported for each factor. Strength of association of each factor with COVID-19 severity was estimated using logistic regression models, overall (whole analysis period) and for the Omicron period. Multivariable analyses including age, sex, and comorbidity count as covariates and were used to determine adjusted odds ratios (ORs) and 95% confidence intervals (95% CIs) for having severe/critical COVID-19 relative to a reference category for each group of factors. For each population and subgroup of HM patients, 30-day and 60-day survival probabilities were estimated using the actuarial survival method, and *p*-values were estimated using the log-rank test within each group of factors. Follow-up time was calculated from time of SARS-CoV-2 diagnosis to time of last hospital visit or death. Kaplan–Meier analyses of overall survival were conducted according to time of diagnosis (pre-Omicron, Omicron BA.1/BA.2, and Omicron BA.4/BA.5 periods) and vaccination status (0, 1–2, and 3–4 COVID-19 vaccine doses) for the overall population and for HM patients hospitalized with COVID-19. Pair-wise comparisons of overall survival were carried out using the log-rank test, *p*-values were adjusted by the Benjamini–Hochberg method, and overall *p*-values were estimated using the log-rank test. Cox proportional-hazard regression models were used to estimate hazard ratios (HRs) and 95% CIs for the COVID-19 risk of death associated with each factor. Adjusted models for each factor included the same three pre-specified variables: age, sex, and comorbidity count. All statistical analyses were generated using R software (version 4.2.2).

## 3. Results

### 3.1. Characteristics of HM Patients with COVID-19 across Time Periods

Of the 32 hospitals affiliated with AHMM, 31 centers covering 98% of the Madrid region population reported 2096 cases of HM patients with COVID-19 to the HEMATO-MADRID COVID-19 registry between 28 February 2020 and 1 October 2022 for possible inclusion in this study (Figure 1). Of these patients, 1818 met the eligibility criteria for this analysis.

The median age of HM patients with COVID-19 included in the present analysis was 70.0 years (IQR 58–78), 57.5% were male, the median number of comorbidities was one (IQR 0–2), and 74.4% and 25.6% had a lymphoid malignancy or a myeloid neoplasia, respectively (Table 1). The most common HM diagnosis was non-Hodgkin lymphoma (NHL), reported in 554/1817 (30.5%) patients, followed by multiple myeloma (MM; 420/1817, 23.1%), chronic lymphocytic leukemia (CLL; 248/1817, 13.6%), acute myeloid leukemia (AML; 148/1817, 8.1%), myelodysplastic syndrome (MDS; 145/1817, 8.0%), and chronic myeloproliferative neoplasm (MPN; 129/1817, 7.0%) (Table 1). Of the 1754 patients with available information, 1228 (70.0%) had received no doses of the COVID-19 vaccine before presenting with COVID-19 disease, 157 (9.0%) had received one or two vaccine doses, and 369 (21.0%) had received three or four doses.

Of the 1818 patients in this analysis, 1281 (70.5%) cases were reported in the pre-Omicron time period, including 1186 (65.2%) and 95 (5.2%) in the D614G- and Alpha/Delta-dominant periods, respectively, and 537 (29.5%) cases were reported in the Omicron time period, including 321 (17.7%) and 216 (11.9%) in the BA.1/BA.2- and BA.4/BA.5-dominant periods, respectively. Table 1 details patient characteristics and COVID-19 management for the pre-Omicron and Omicron periods. In the Omicron period, the median age, percentage of patients who were male, and number of comorbidities was lower, and the percentage of patients with a lymphoid malignancy and active cancer therapy was higher, compared with the pre-Omicron period. Reflecting the COVID-19 vaccination roll-out over time, 73/1273 (5.7%) patients had been vaccinated with ≥1 dose among cases reported in the pre-Omicron time period compared to 453/481 (94.2%) cases reported in the Omicron time period. Similarly, the pharmacologic therapies administered for COVID-19 differed between time periods, reflecting the introduction over time of remdesivir, nirmatrelvir/ritonavir, and monoclonal antibodies and the reduction in the use of tocilizumab.

### 3.2. Factors Associated with COVID-19 Severity

Data on the clinical severity of COVID-19 were available for 1781/1818 patients (98.0%), of whom 1020 (57.3%) had mild/moderate disease and 761 (42.7%) had severe/critical COVID-19. The proportion of cases that were severe/critical COVID-19 decreased from 53.5% to 17.6% and the proportion of patients hospitalized decreased from 75.3% to 35.7% in the pre-Omicron and Omicron time periods, respectively (Table 1).

Patients with severe/critical COVID-19 included a higher proportion aged ≥60 years (634/761, 83.3%, vs 647/1019, 63.5%; *p* < 0.001) and a higher proportion with ≥2 comorbidities (260/761, 34.2%, vs 192/1020, 18.8%; *p* < 0.0001) compared to the mild/moderate group (Appendix A). After adjusting for age, sex, and comorbidities, the following baseline characteristics and cancer features were independently associated with severe/critical COVID-19 disease: patients aged ≥70 years (OR 2.16, 95% CI 1.64–2.87), with >1 comorbidity (2.44, 1.85–3.21), or with an underlying HM of CLL (1.64, 1.19–2.27), had greater odds of severe/critical COVID-19, whereas the odds were lower among patients who had COVID-19 during the BA.1/BA.2 (0.28, 0.2–0.37) or BA.4/BA.5 (0.13, 0.08–0.19) time periods, and among those who had received one or two (0.51, 0.34–0.75) or three or four (0.22, 0.16–0.29) COVID-19 vaccine doses. Overall, patients on active antineoplastic therapy (0.69, 0.56–0.86), and particularly those receiving immunotherapy (0.47, 0.33–0.67), as well as patients who had undergone autologous hematopoietic stem cell transplantation (0.68, 0.46–0.97), had lower odds of severe/critical COVID-19 (Appendix A).

Focusing on cases diagnosed within the Omicron era, patients with severe/critical COVID-19 were more commonly aged ≥60 years (79/94, 84.0%, vs. 281/441, 63.7%; *p* < 0.001), male (62/94, 66.09%, vs 222/440, 50.3%; *p* < 0.007), and had >1 comorbidities (22/94, 23.4%, vs 59/441, 13.4%; *p* < 0.002) compared to the mild/moderate group (Appendix A). After adjusting for age, sex, and comorbidities, the odds of having severe/critical disease were higher in patients aged ≥70 years (OR 2.53, 95% CI 1.28–5.19) and lower in females versus males (0.58, 0.36–0.93), in patients diagnosed during the Omicron BA.4/BA.5 vs. BA.1/BA.2 time period (0.46, 0.25–0.75), and in patients with MM (0.30, 0.15–0.58), AML (0.37, 0.13–0.89), or MPN (0.18, 0.03–0.66) (Appendix A).

### 3.3. Factors Associated with COVID-19 Hospitalization

Data on the care setting of COVID-19 treatment were available for 1814/1818 HM patients (99.8%), of whom 1154 (63.6%) were hospitalized and received treatment as inpatients and 660 (36.4%) received treatment in the outpatient setting. Overall, the hospitalization rate decreased from 963/1279 (75.3%) in the pre-Omicron time period to 191/535 (35.7%) during the Omicron time period, as did the proportion of hospitalized patients who were admitted to the intensive care unit (289/963 [30.0%] to 28/191 [14.7%]; Table 1). Hospitalization rates gradually reduced over time, from 76.5% (907/1185) and 59.6% (56/94) during the D614G- and Alpha/Delta-dominant periods, respectively, to 41.1% (132/321) and 27.6% (59/214) during the Omicron BA.1/BA.2- and BA.4/BA.5-dominant periods, respectively (*p* < 0.001). The rate of ICU admission similarly decreased over time periods, with respective rates of 23.0% (273/1185), 17.0% (16/94), 7.2% (23/321), and 2.3% (5/214) (*p* < 0.0001). After adjusting for age, sex, and comorbidities, patients diagnosed during the Omicron period had lower odds of hospitalization than those diagnosed during the pre-Omicron period (OR 0.20, 95% CI 0.16–0.25), and among inpatients the odds of requiring ICU admission were lower in the Omicron period (0.41, 0.26–0.62).

### 3.4. Factors Associated with COVID-19 Mortality

After a median follow-up of 54 days (IQR 20–147), 462/1817 (25.4%) HM patients with COVID-19 had died; the mortality rate was 31.9% (409/1281) in the pre-Omicron time period and 9.9% (53/536) in the Omicron time period (Figure 2). In the pre-Omicron time period, the COVID-19 mortality risk was greater in patients aged ≥70 years (hazard ratio [HR] 2.57, 95% CI 2.03–3.25), those with ≥2 comorbidities (1.53, 1.15–2.04), and patients receiving conventional chemotherapy (1.35, 1.05–1.74), and lower in those who had received one or two (0.52, 0.30–0.90) or three or four (0.15, 0.04–0.62) vaccine doses. In the Omicron time period, an age ≥70 years (HR 3.19, 95% CI 1.59–6.42) and receipt of allogeneic stem cell transplantation (3.12, 1.17–8.36) were associated with higher mortality risk (Figure 2).

Kaplan–Meier analyses of survival are shown in Figure 3. Overall and time period-specific 30-day and 60-day survival estimates are summarized in Appendix A for all patients and for subgroups, including by malignancy type, cancer therapy, transplantation, and COVID-19 treatment. Among all patients, survival probabilities were 78% (95% CI 76–80) at 30 days and 70% (67–72) at 60 days (Appendix A). Kaplan–Meier analysis demonstrated significantly better survival among HM patients diagnosed with COVID-19 in the Omicron BA.4/BA.5 or Omicron BA.1/BA.2 periods compared with the pre-Omicron period (both *p* < 0.0001; Figure 3A). Vaccination status also had a significant impact; overall, HM patients who had received three or four or one or two doses of the COVID-19 vaccine had significantly better survival than unvaccinated patients (both *p* < 0.0001; Figure 3B), and findings were similar for the pre-Omicron period but with *p*-values of 0.002 and 0.003 for the comparison of patients receiving three or four or one or two doses versus unvaccinated patients, respectively (Figure 3C). No difference in survival was seen between the three groups during the Omicron period (Figure 3D).

### 3.5. Factors Associated with COVID-19 Mortality in Hospitalized HM Patients

Of the 1154 patients who were hospitalized, 440 (38.1%) died, with mortality rates of 41.3% (398/963) and 22.0% (42/191) in the pre-Omicron and Omicron time periods, respectively (Figure 4), after respective median observation periods of 50 days (IQR 23–161) and 85 days (30–152). In the pre-Omicron time period, the COVID-19 mortality risk was greater in hospitalized HM patients aged ≥70 years (HR 2.13, 95% CI 1.68–2.70), with ≥2 comorbidities (1.43, 1.07–1.90), or receiving conventional chemotherapy (1.33, 1.03–1.72), and lower in patients who had received three or four vaccine doses (0.20, 0.05–0.79) (Figure 4). In the Omicron time period, no factors demonstrated significant associations with mortality risk, with wide 95% CIs all overlapping one (Figure 4).

Kaplan–Meier analyses of survival are shown in Figure 5. Overall and time period-specific 30-day and 60-day survival estimates are summarized in Table 2 for all patients and by subgroup. Among all patients, survival probabilities were 67% (95% CI 65–70) at 30 days and 57% (53–60) at 60 days (Table 2). Kaplan–Meier analysis demonstrated significantly better survival among HM patients diagnosed with COVID-19 in the Omicron BA.4/BA.5 (*p* < 0.0001) or Omicron BA.1/BA.2 (*p* = 0.003) periods compared with the pre-Omicron period (Figure 5A). Vaccination status also significantly impacted survival; overall, hospitalized HM patients who had received three or four (*p* < 0.0001) or one or two (*p* = 0.0005) doses of COVID-19 vaccine had significantly better survival than unvaccinated patients (Figure 5B), and similar findings were seen for the pre-Omicron period but with respective *p*-values of 0.03 and 0.05 (Figure 5C). No difference in survival of hospitalized HM patients was seen between the three groups during the Omicron period (Figure 5D).

Actuarial 30-day and 60-day survival rates showed that, among all hospitalized HM patients, an older age (*p* < 0.001), a greater number of comorbidities (*p* < 0.001), and treatment with corticosteroids (*p* < 0.001) were significantly associated with lower survival rates (Table 2). Conversely, having undergone stem cell transplantation (*p* < 0.001), having received COVID-19 vaccination(s) (*p* < 0.001), and treatment with remdesivir (*p* < 0.001) or monoclonal antibodies (*p* = 0.02) were associated with greater survival rates. During the pre-Omicron time period, similar associations with survival rates were seen for age, comorbidities, stem cell transplantation, COVID-19 vaccination, and treatment with remdesivir or corticosteroids, whereas during the Omicron time period none of the factors analyzed were associated with survival rates (Table 2).

## 4. Discussion

The findings from this large registry-based study demonstrate that COVID-19 outcomes among HM patients have considerably improved over the course of the pandemic, with the hospitalization rate having fallen from 75.3% in the pre-Omicron time period to 35.7% in the Omicron era and the mortality rate having fallen from 31.9% to 9.9%. However, the mortality remains high in HM patients hospitalized with COVID-19, at 22.0%. We found that HM patients diagnosed with COVID-19 during the Omicron period had five-fold lower odds of having severe/critical COVID-19 (OR 0.21), a 52.6% lower risk of hospitalization, and a 63.9% lower risk of 30-day overall mortality than those diagnosed with COVID-19 during the pre-Omicron period. Among the factors related to this improvement are the lower disease severity associated with the new Omicron variants [33], the roll-out of intensive COVID-19 vaccination programs, and the introduction of more effective COVID-19 therapies such as nirmatrelvir/ritonavir, remdesivir, and monoclonal antibodies.

As of the end of the first quarter of 2023, the trajectory of the COVD-19 pandemic remains unclear, particularly in HM patients. Our current analysis shows that many of the risk factors that were strongly associated with COVID-19 mortality in the early phase of the pandemic now have an attenuated or no association with mortality during the Omicron-dominant period. During the pre-Omicron period, we found that an age ≥70 years, the presence of >1 comorbidity, and receiving active conventional chemotherapy were associated with higher COVID-19 mortality risk in all HM patients and in hospitalized HM patients, consistent with our previous report [5] and other studies [2,3,4,6,8], whereas having received a primary-series vaccination and at least one booster was associated with lower in-hospital mortality risk. In contrast, during the Omicron time period, our multivariable analysis showed that, of these factors, the only one remaining independently related to the risk of death was age ≥70 years. This finding is in line with two reports from the EPICOVIDEHA survey in HM patients with COVID-19 due to the initial Omicron variant and subvariants, in which only advanced age and active cancer were associated with higher mortality on univariable analysis [26,27]. Taken together, these findings underline the changes seen over time in the risk factors associated with COVID-19 mortality among HM patients during the Omicron-dominant period. However, it is unclear whether these changes are due to primary-series and booster COVID-19 vaccinations, changes in the propensity of the virus to cause severe disease, improvements in disease management, or changes in the clinical profile of HM patients with COVID-19.

With regard to COVID-19 management, our findings show that the use of corticosteroids was associated with an increased mortality risk overall and in the pre-Omicron and Omicron time periods, confirming our previous results and data from EPICOVIDEHA [34,35]. Based on this robust evidence, we recommend avoiding the use of corticosteroids in HM patients with COVID-19 [36]. Similarly, in our series tocilizumab treatment did not improve COVID-19 outcomes in HM patients, and we therefore do not recommend its use in this setting. In contrast, treatment with remdesivir and monoclonal antibodies in hospitalized HM patients, as well as with nirmatrelvir/ritonavir in the overall population, was associated with a reduced risk of death, in line with what was reported in the EPICOVIDEHA study [34]. Although data from well-designed clinical trials, specifically in HM patients, are not currently available, our findings and those from EPICOVIDEHA suggest that these are reasonable therapeutic strategies in high-risk HM patients.

Among the strengths of this study, which represents our first analysis of COVID-19 severity and mortality among HM patients in the Omicron era, are its prospective, comprehensive collection of clinical and outcome data on HM patients with COVID-19, the use of multivariable analysis to identify independent risk factors for COVID-19 mortality, the long period covered, including the length of follow-up, and the fact that our patient series is highly representative of this population. A limitation of our study is that it is based on registry data. Although to the best of our knowledge the registry includes all HM patients with COVID-19, the true patient population may be higher because of low rates of testing or misdiagnoses in the first period of the pandemic. Treating physicians established close contact with their patients and special access paths to facilitate the care and inclusion of virtually all patients with hematologic neoplasms, particularly those under active treatment in the registry. Another limitation is that our case series incorporates a heterogeneous patient population with multiple different HM; nevertheless, the size of the population, including the large numbers of patients with specific malignancies (more than 100 patients in six out of nine HM), the detailed reporting by HM, and the long follow-up, could mitigate in part this perceived limitation.

## 5. Conclusions

This study provides a rare and valuable framework to show strong evidence of change in the clinical picture and mid-term outcomes over more than two years of the COVID-19 pandemic across main subtypes of hematological malignancies. COVID-19 mortality in HM patients has decreased considerably in the Omicron period of the pandemic, and the clinical management of patients with COVID-19 has improved thanks to the addition of new antiviral therapies and monoclonal antibodies. However, mortality in hospitalized HM patients remains high. We suggest that specific studies of novel COVID-19 therapies in immunocompromised patients should be undertaken with the aim of further improving outcomes, and that HM patients should receive active protection against SARS CoV-2 infection and severe outcomes through vaccination and preventive interventions.

## Figures and Tables

**Figure 1 cancers-16-00379-f001:**
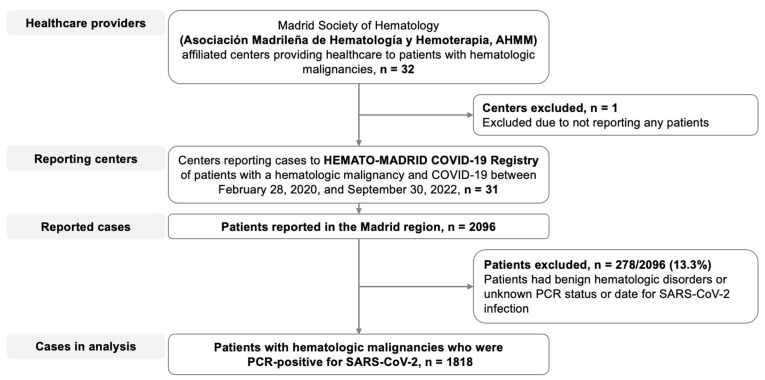
Flow diagram. Patients with hematologic malignancies who were reported as having COVID-19 and who were included in the present analysis.

**Figure 2 cancers-16-00379-f002:**
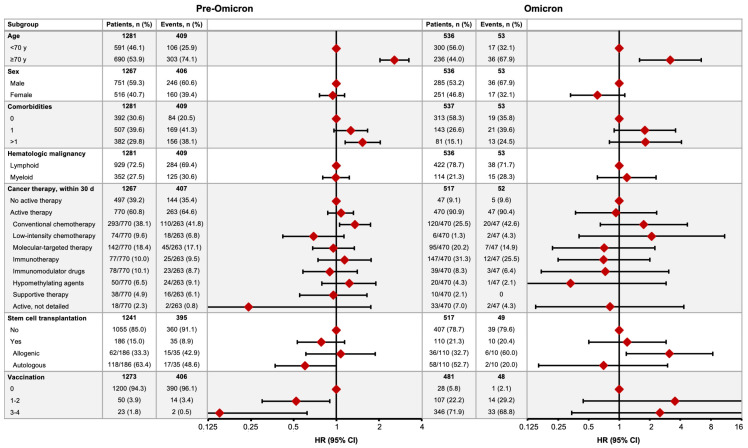
COVID-19 mortality in the pre-Omicron and Omicron time periods. Figures show the numbers of patients who died (no. of events) by subgroup and the relative hazard ratio (HR) for COVID-19 mortality between associated subgroups (HR = 1 for reference subgroup in each set).

**Figure 3 cancers-16-00379-f003:**
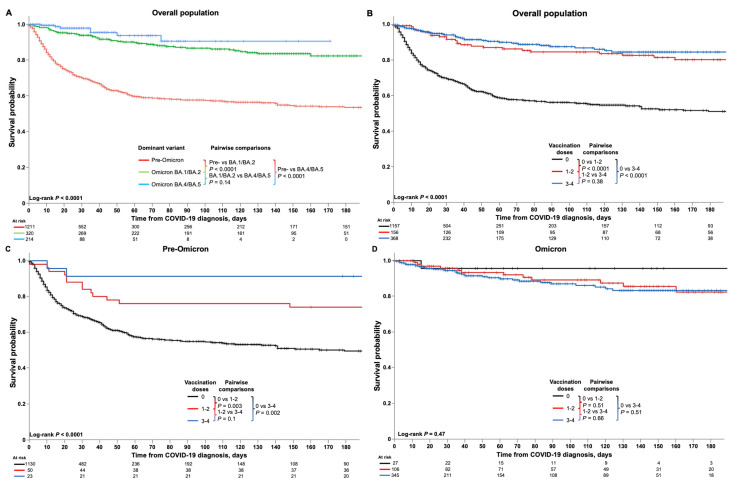
Kaplan–Meier analyses of survival outcomes in HM patients with COVID-19. Figures show the survival estimates among HM patients with COVID-19 (**A**) according to time period (pre-Omicron, Omicron BA.1/BA.2, and Omicron BA.4/BA.5), and (**B**) overall and (**C**,**D**) in the pre-Omicron and Omicron time periods among patients who were unvaccinated or who had received one or two or three or four vaccinations at the time of their COVID-19 diagnosis. Analyses of 30-day and 60-day survival rates showed that, among all HM patients, older age (*p* < 0.001), male sex (*p* = 0.04), greater number of comorbidities (*p* < 0.001), and treatment with tocilizumab (*p* < 0.001) or corticosteroids (*p* < 0.001) were significantly associated with lower survival rates (Appendix A). Conversely, having undergone stem cell transplantation (*p* < 0.001), having received COVID-19 vaccination(s) (*p* < 0.001), and treatment with nirmatrelvir/ritonavir (*p* < 0.001) or remdesivir (*p* < 0.03) were associated with greater survival rates. Additionally, specific HM including CML and MPN, and having received active cancer therapy, notably with immunotherapy, were also associated with greater survival rates. During the pre-Omicron time period, similar associations with survival rates were seen for age, comorbidities, stem cell transplantation, COVID-19 vaccination, and treatment with corticosteroids, whereas during the Omicron time period, survival rates were associated with age, sex, comorbidities, and tocilizumab or corticosteroid treatment. Notably, remdesivir treatment during the Omicron time period was associated with lower survival rates, in contrast to the overall findings (Appendix A).

**Figure 4 cancers-16-00379-f004:**
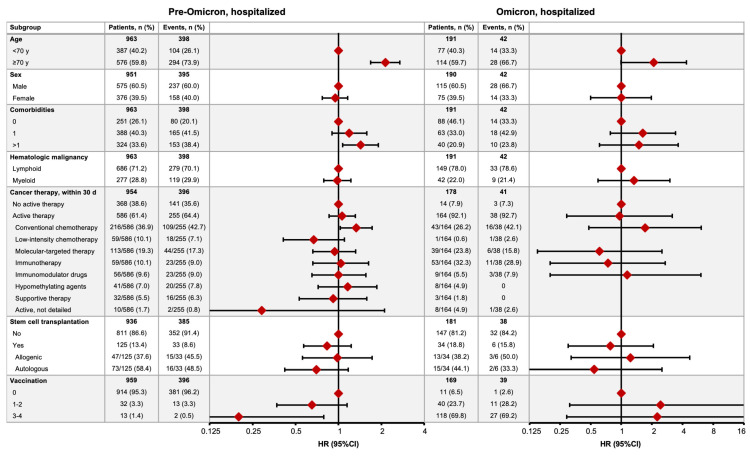
COVID-19 mortality in hospitalized HM patients in the pre-Omicron and Omicron time periods. Figures show the numbers of patients who died (no. of events) by subgroup and the relative hazard ratio (HR) for COVID-19 mortality between associated subgroups (HR = 1 for reference subgroup in each set).

**Figure 5 cancers-16-00379-f005:**
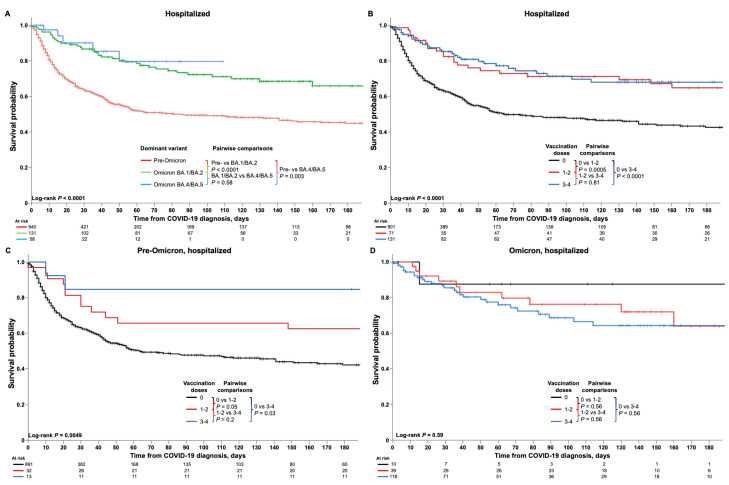
Kaplan–Meier analyses of survival outcomes in HM patients hospitalized due to COVID-19. Figures show the survival estimates among HM patients hospitalized due to COVID-19 (**A**) according to time period (pre-Omicron, Omicron BA.1/BA.2, and Omicron BA.4/BA.5), and (**B**) overall and (**C**,**D**) in the pre-Omicron and Omicron time periods among patients who were unvaccinated or who had received one or two or three or four vaccinations at the time of their COVID-19 diagnosis.

**Table 1 cancers-16-00379-t001:** Baseline characteristics of, and therapy received by, patients with hematologic malignancies and COVID-19.

	Overall Population, *n* = 1818			Inpatient Population, *n* = 1154		
	Time Period (by Dominant SARS-CoV-2 Variant)			Time Period (by Dominant SARS-CoV-2 Variant)		
	Pre-Omicron, *n* = 1281	Omicron, *n* = 537	*p*-Value *	Pre-Omicron, *n* = 963	Omicron, *n* = 191	*p*-Value *
Age, y	*n* = 1281	*n* = 536		*n* = 963	*n* = 191	
Median (IQR)	70.0 (59–79)	67.0 (55–77)	**<0.001**	72.0 (62–80)	73.0 (64–80)	0.7
Age <60 y, *n* (%)	333 (26.0)	175 (32.6)		197 (20.5)	34 (17.8)	
Age 60–70 y, *n* (%)	258 (20.1)	125 (23.3)		190 (19.7)	43 (22.5)	
Age 70–80 y, *n* (%)	398 (31.1)	152 (28.4)		325 (33.7)	64 (33.5)	
Age >80 y, *n* (%)	292 (22.8)	84 (15.7)		251 (26.1)	50 (26.2)	
**Sex, *n* (%)**	***n*** **= 1267**	***n*** **= 536**		***n*** **= 951**	***n*** **= 190**	
Male	751 (59.3)	285 (53.2)	**0.017**	575 (60.5)	115 (60.5)	>0.9
Female	516 (40.7)	251 (46.8)		376 (39.5)	75 (39.5)	
**Comorbidities, *n* (%)**	***n*** **= 1281**	***n*** **= 537**		***n*** **= 963**	***n*** **= 191**	
0	392 (30.6)	313 (58.3)	**<0.001**	251 (26.1)	88 (46.1)	**<0.001**
1	507 (39.6)	143 (26.6)		388 (40.3)	63 (33.0)	
>1	382 (29.8)	81 (15.1)		324 (33.6)	40 (20.9)	
**Hematologic malignancy, *n* (%)**	***n*** **= 1281**	***n*** **= 536**		***n*** **= 963**	***n*** **= 191**	
**Lymphoid malignancy**	929 (72.5)	422 (78.7)	**0.006**	686 (71.2)	149 (78.0)	0.057
**Type of lymphoid malignancy**	***n*** **= 929**	***n*** **= 422**		***n*** **= 686**	***n*** **= 149**	
NHL	366 (39.4)	188 (44.5)		277 (40.4)	84 (56.4)	
ALL	29 (3.1)	24 (5.7)		20 (2.9)	6 (4.0)	
CLL	187 (20.1)	61 (14.5)		148 (21.6)	28 (18.8)	
HL	54 (5.8)	22 (5.2)		34 (5.0)	0 (0)	
MM	293 (31.5)	127 (30.1)		207 (30.2)	31 (20.8)	
**Myeloid malignancy**	352 (27.5)	114 (21.3)		277 (28.8)	42 (22.0)	
**Type of myeloid malignancy**	***n*** **= 352**	***n*** **= 114**		***n*** **= 277**	***n*** **= 42**	
AML	97 (27.6)	51 (44.7)		77 (27.8)	24 (57.1)	
CML	36 (10.2)	8 (7.0)		20 (7.2)	2 (4.8)	
MDS	118 (33.5)	27 (23.7)		103 (37.2)	11 (26.2)	
MPN	101 (28.7)	28 (24.6)		77 (27.8)	5 (11.9)	
**Cancer therapy, within 30 d, *n* (%)**	***n*** **= 1267**	***n*** **= 517**		***n*** **= 954**	***n*** **= 178**	
No active therapy	497 (39.2)	47 (9.1)	**<0.001**	368 (38.6)	14 (7.9)	**<0.001**
Active therapy	770 (60.8)	470 (90.9)		586 (61.4)	164 (92.1)	
**Type of active therapy**	***n*** **= 770**	***n*** **= 470**		***n*** **= 586**	***n*** **= 164**	
Conventional chemotherapy	293 (38.1)	120 (25.5)		216 (36.9)	43 (26.2)	
Low-intensity chemotherapy	74 (9.6)	6 (1.3)		59 (10.1)	1 (0.6)	
Molecular-targeted therapy	142 (18.4)	95 (20.2)		113 (19.3)	39 (23.8)	
Immunotherapy	77 (10.0)	147 (31.3)		59 (10.1)	53 (32.3)	
Immunomodulatory drugs	78 (10.1)	39 (8.3)		56 (9.6)	9 (5.5)	
Hypomethylating agents	50 (6.5)	20 (4.3)		41 (7.0)	8 (4.9)	
Supportive therapy	38 (4.9)	10 (2.1)		32 (5.5)	3 (1.8)	
Active, not detailed	18 (2.3)	33 (7.0)		10 (1.7)	8 (4.9)	
**Cellular therapy, *n* (%)**	***n*** **= 1241**	***n*** **= 517**		***n*** **= 936**	***n*** **= 181**	
No	1055 (85.0)	407 (78.7)	**0.001**	811 (86.6)	147 (81.2)	0.057
Yes	186 (15.0)	110 (21.3)		125 (13.4)	34 (18.8)	
**Type of cellular therapy**	***n*** **= 186**	***n*** **= 110**		***n*** **= 125**	***n*** **= 34**	
Allogenic	62 (33.3)	36 (32.7)		47 (37.6)	13 (38.2)	
Autologous	118 (63.4)	58 (52.7)		73 (58.4)	15 (44.1)	
CAR T cell	6 (3.2)	16 (14.5)		5 (4.0)	6 (17.6)	
**Vaccination, *n* (%)**	***n*** **= 1273**	***n*** **= 481**		***n*** **= 959**	***n*** **= 169**	
0	1200 (94.3)	28 (5.8)	**<0.001**	914 (95.3)	11 (6.5)	**<0.001**
1–2	50 (3.9)	107 (22.2)		32 (3.3)	40 (23.7)	
3–4	23 (1.8)	346 (71.9)		13 (1.4)	118 (69.8)	
**Pharmacologic therapies for COVID-19, *n* (%)**	***n*** **= 1281**	***n*** **= 537**		***n*** **= 963**	***n*** **= 191**	
Nirmatrelvir/ritonavir	0 (0)	97 (18.1)	>0.9	0 (0)	10 (5.2)	>0.9
Remdesivir	94 (7.3)	159 (29.6)	**<0.001**	94 (9.8)	104 (54.5)	**<0.001**
Tocilizumab	195 (15.2)	25 (4.7)	**<0.001**	195 (20.2)	25 (13.1)	**0.02**
Monoclonal antibodies	2 (0.2)	32 (6.0)	**<0.001**	2 (0.2)	27 (14.1)	**<0.001**
Corticosteroids	680 (53.1)	119 (22.2)	**<0.001**	633 (65.7)	114 (59.7)	0.1
**Care setting of COVID-19 treatment, *n***(%)	***n*** **= 1279**	***n*** **= 535**		***n*** **= 963**	***n*** **= 191**	
Outpatient	316 (24.7)	344 (64.3)	**<0.001**	0 (0)	0 (0)	n/a
Hospitalized	963 (75.3)	191 (35.7)	**<0.001**	963 (100)	191 (100)	**<0.001**
Intensive care unit	289 (22.6)	28 (5.2)		289 (30.0)	28 (14.7)	
**Maximum clinical severity of COVID-19, *n* (%)**	***n*** **= 1246**	***n*** **= 535**		***n*** **= 963**	***n*** **= 191**	
Mild/Moderate	579 (46.5)	441 (82.4)	**<0.001**	286 (29.7)	100 (52.4)	**<0.001**
Severe/Critical	667 (53.5)	94 (17.6)		667 (69.3)	91 (47.6)	

ALL, acute lymphoid leukemia; AML, acute myeloid leukemia; CAR, chimeric antigen receptor; CLL, chronic lymphocytic leukemia; CML, chronic myeloid leukemia; COVID-19, coronavirus disease 2019; HL, Hodgkin lymphoma; IQR, interquartile range; MDS, myelodysplastic syndrome; MPN, myeloproliferative neoplasm; MM, multiple myeloma; n/a, not applicable; NHL, non-Hodgkin lymphoma; SARS-CoV-2, severe acute respiratory syndrome coronavirus-2; y, years. * *p*-values for comparisons between pre-Omicron and Omicron subgroups estimated using the Wilcoxon test for age (median), the Cochran–Armitage test for comorbidities and vaccination, and univariate logistic regression for the rest of the clinical variables.

**Table 2 cancers-16-00379-t002:** Actuarial 30-day and 60-day survival in hospitalized patients with hematologic malignancies and COVID-19, overall and in pre-Omicron and Omicron time periods.

	Overall Population			Pre-Omicron Time Period			Omicron Time Period		
	Survival Estimate, % (95% CI)		*p*-Value *	Survival Estimate, % (95% CI)		*p*-Value *	Survival Estimate, % (95% CI)		*p*-Value *
	30 Days	60 Days		30 Days	60 Days		30 Days	60 Days	
**Overall**	**67 (65–70)**	**57 (53–60)**		64 (60–67)	52 (48–56)		87 (82–93)	79 (72–86)	
**Age**			**<0.001**			**<0.001**			0.3
<60 y	87 (82–92)	76 (69–83)		85 (80–91)	74 (67–82)		96 (89–100)	88 (76–100)	
60–70 y	77 (72–83)	66 (59–73)		73 (66–80)	61 (53–70)		95 (88–100)	83 (71–96)	
70–80 y	66 (61–71)	54 (48–60)		62 (56–68)	48 (42–56)		84 (74–94)	77 (66–89)	
>80 y	48 (42–55)	39 (33–46)		43 (37–50)	33 (27–41)		78 (66–93)	71 (57–88)	
**Sex**			0.7			0.7			0.7
Male	67 (63–71)	56 (51–60)		63 (59–68)	51 (46–56)		86 (79–93)	77 (69–86)	
Female	68 (64–73)	58 (53–64)		64 (59–69)	53 (48–60)		90 (83–98)	81 (71–93)	
**Comorbidities**			**<0.001**			**<0.001**			0.1
0	79 (75–84)	70 (64–76)		74 (69–81)	65 (58–72)		93 (87–99)	83 (75–93)	
1	67 (63–72)	54 (49–60)		66 (61–71)	51 (45–57)		78 (67–90)	73 (62–86)	
>1	57 (51–63)	47 (41–53)		53 (47–59)	43 (37–50)		91 (81–100)	78 (64–96)	
**Hematologic malignancy**			0.2			0.4			0.8
** Lymphoid malignancy **	67 (64–71)	57 (53–61)		63 (59–67)	51 (47–56)		88 (82–94)	80 (73–88)	
NHL	71 (66–76)	60 (54–66)		65 (59–72)	53 (46–60)		87 (79–95)	80 (72–90)	
ALL	82 (67–100)	45 (25–80)		76 (58–100)	48 (27–87)		100 (100–100)	50 (19–100)	
CLL	61 (54–70)	56 (48–65)		56 (48–65)	49 (40–60)		91 (80–100)	91 (80–100)	
HL	67 (53–86)	63 (47–83)		67 (53–86)	63 (47–83)		NE (NE-NE)	NE (NE-NE)	
MM	66 (60–73)	54 (47–62)		63 (57–71)	51 (43–59)		85 (73–100)	75 (59–95)	
** Myeloid malignancy **	68 (62–73)	56 (50–63)		65 (59–71)	53 (47–61)		86 (75–100)	74 (60–93)	
AML	65 (56–76)	57 (48–69)		59 (48–72)	50 (39–64)		89 (76–100)	83 (68–100)	
CML	84 (69–100)	84 (69–100)		82 (66–100)	82 (66–100)		100 (100–100)	100 (100–100)	
MDS	58 (48–68)	45 (35–58)		56 (47–68)	44 (34–58)		75 (50–100)	56 (28–100)	
MPN	82 (73–92)	62 (49–79)		81 (72–92)	64 (51–81)		100 (100–100)	NE (NE-NE)	
**Cancer therapy, within 30 d**			0.2			>0.9			0.8
No active therapy	63 (58–68)	53 (47–60)		62 (56–67)	52 (46–59)		100 (100–100)	88 (67–100)	
** Active therapy **	69 (65–73)	58 (54–62)		64 (60–69)	52 (47–57)		86 (80–92)	78 (71–85)	
Conventional chemotherapy	63 (57–69)	49 (42–56)		61 (54–68)	46 (39–55)		75 (62–90)	61 (47–80)	
Low-intensity chemotherapy	72 (61–86)	55 (39–76)		74 (62–88)	56 (40–78)		NE (NE-NE)	NE (NE-NE)	
Molecular-targeted therapy	73 (65–81)	65 (57–75)		67 (58–77)	60 (50–72)		90 (79–100)	81 (68–98)	
Immunotherapy	81 (73–89)	72 (64–82)		68 (57–82)	56 (44–72)		94 (87–100)	89 (81–99)	
Immunomodulator drugs	62 (50–76)	51 (39–68)		62 (50–77)	49 (35–68)		58 (31–100)	58 (31–100)	
Hypomethylating agents	66 (53–82)	62 (48–80)		60 (45–79)	54 (39–76)		100 (100–100)	100 (100–100)	
Supportive therapy	62 (47–83)	43 (27–69)		61 (46–82)	43 (27–68)		NE (NE-NE)	NE (NE-NE)	
Active, not detailed	92 (79–100)	69 (44–100)		89 (71–100)	76 (52–100)		100 (100–100)	67 (30–100)	
**Cellular therapy**			**<0.001**			**<0.001**			0.2
No	64 (61–68)	54 (51–58)		61 (57–64)	50 (46–54)		86 (80–92)	79 (71–87)	
** **Yes** **	85 (79–91)	72 (64–81)		82 (74–90)	69 (59–80)		97 (91–100)	84 (71–100)	
Allogenic	89 (81–98)	72 (59–86)		86 (76–97)	70 (56–87)		100 (100–100)	80 (59–100)	
Autologous	80 (72–90)	74 (64–86)		77 (67–89)	72 (60–86)		93 (80–100)	84 (65–100)	
CAR T cell	100 (100–100)	60 (29–100)		100 (100–100)	NE (NE-NE)		100 (100–100)	100 (100–100)	
**Vaccinations**			**<0.001**			**0.003**			0.6
0	63 (60–67)	51 (47–55)		63 (60–67)	51 (47–55)		88 (67–100)	88 (67–100)	
1–2	82 (73–92)	74 (64–86)		74 (60–91)	65 (50–84)		89 (80–100)	83 (71–96)	
3–4	85 (79–92)	77 (69–86)		85 (67–100)	85 (67–100)		85 (79–93)	76 (67–86)	
**COVID-19 therapies**									
**Nirmatrelvir/ritonavir**			0.3			n/a			0.8
No	67 (64–70)	56 (53–60)		NE (NE-NE)	NE (NE-NE)		87 (82–93)	79 (73–86)	
Yes	89 (71–100)	71 (43–100)		NE (NE-NE)	NE (NE-NE)		89 (71–100)	71 (43–100)	
**Remdesivir**			**<0.001**			**0.002**			0.6
No	64 (60–67)	54 (50–58)		62 (58–65)	51 (47–55)		83 (75–92)	80 (71–89)	
Yes	87 (82–92)	71 (64–80)		82 (73–91)	65 (54–77)		92 (86–98)	78 (69–88)	
**Tocilizumab**			0.5			0.14			0.3
No	67 (64–71)	56 (53–60)		63 (59–66)	51 (47–56)		89 (84–94)	80 (73–87)	
Yes	69 (63–76)	58 (51–65)		68 (61–75)	55 (48–64)		78 (63–97)	73 (57–94)	
**Monoclonal antibodies**			**0.02**			n/a			0.7
No	67 (64–70)	56 (53–60)		NE (NE-NE)	NE (NE-NE)		87 (81–93)	80 (73–87)	
Yes	92 (81–100)	77 (61–97)		NE (NE-NE)	NE (NE-NE)		92 (81–100)	77 (61–97)	
**Corticosteroids**			**<0.001**			**<0.001**			0.2
No	76 (72–81)	68 (63–73)		72 (67–78)	64 (58–71)		92 (86–99)	83 (73–93)	
Yes	63 (59–67)	51 (47–55)		60 (56–64)	46 (42–51)		84 (77–92)	76 (68–86)	

ALL, acute lymphoid leukemia; AML, acute myeloid leukemia; CAR, chimeric antigen receptor; CI, confidence interval; CLL, chronic lymphocytic leukemia; CML, chronic myeloid leukemia; COVID-19, coronavirus disease 2019; HL, Hodgkin lymphoma; IQR, interquartile range; MDS, myelodysplastic syndrome; MPN, myeloproliferative neoplasm; MM, multiple myeloma; n/a, not applicable; NE, not estimable; NHL, non-Hodgkin lymphoma; SARS-CoV-2, severe acute respiratory syndrome coronavirus-2; y, years. * *p*-values estimated using log-rank test.

## Data Availability

The datasets generated during and/or analyzed during the current study are available from the corresponding author on reasonable request.

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
