# Peer review of "COVID-19 Outcomes in Patients with Hematologic Malignancies in the Era of COVID-19 Vaccination and the Omicron Variant"

_cancers, 2024, doi:10.3390/cancers16020379_

Round 1
Reviewer 1 Report
Comments and Suggestions for Authors
The manuscript explores the evolving landscape of COVID-19 outcomes in hematologic malignancy patients, particularly during the Omicron era, highlighting a significant decrease in mortality but emphasizing persistent challenges, such as high mortality in hospitalized patients. Very interesting to read, but a few comments for the author's attention are listed below:
Comments for authors:
Title:
The title accurately reflects the content of the study and is concise and informative.
Introduction:
1. The section mentions mortality rates in the pre-vaccination era (13–37% overall, 19–46% for hospitalized patients) and briefly refers to rates in studies during different dominant periods. However, a more concise and explicit presentation of mortality rates across these periods would enhance clarity.
2. While the section mentions three real-world studies on COVID-19 among HM patients, it could benefit from a brief synthesis or comparison of findings from these studies. This would strengthen the narrative and provide a clearer picture of the existing evidence.
3. Given the evolving landscape and the focus on Omicron, the section could provide more detailed insights into the impact of this variant on HM patients. This could include specific findings from the third study conducted during the Omicron-dominant period.
Methods:
1. The section mentions that there were no specified follow-up time-points for patients added to the database. Providing a brief explanation or rationale for this decision would enhance understanding.
2. While it is mentioned that disease severity was assessed per World Health Organization guidelines, a brief overview or reference to these guidelines could aid readers in understanding the criteria used for severity assessment.
3. Although the waiver of written informed consent is mentioned, providing a concise rationale or reference for this decision would strengthen the ethical transparency of the study.
4. While the study design and participant information are briefly mentioned, more detailed information regarding the inclusion and exclusion criteria, as well as the rationale behind the chosen periods for analysis, would enhance the methodological transparency
Results:
1. It would be beneficial to provide a concise overview of the methodology used for the Kaplan–Meier analyses. Clarifying how data were collected, inclusion criteria, and statistical methods used would strengthen the scientific foundation of the study.
2. While the section delves into detailed analyses, there is a potential improvement in introducing the background and context more explicitly. A brief overview or rationale for studying survival outcomes in HM patients with COVID-19 could enhance the introduction.
3. The statistical methods used in the analysis are adequately described, but further elaboration on the rationale for choosing specific statistical tests and adjustments would enhance the robustness of the findings.
Discussion:
1. he study's limitation regarding the heterogeneity of the patient population is acknowledged. While the large sample size mitigates this to some extent, it raises questions about the generalizability of findings to specific HM subgroups.
2. A notable limitation is the reliance on registry data. While efforts are made to include all HM patients with COVID-19, the study acknowledges potential underrepresentation due to low testing rates or misdiagnoses in the early stages of the pandemic.
Author Response
I attach a Word containing reviewers replies.

Reviewer 2 Report
Comments and Suggestions for Authors
The present paper highlights the implications of COVID-19 outcomes in patients with hematologic malignancies in the era of COVID-19 vaccination and the omicron variant in a comprehensive study conducted in 31 centers. The topic is relevant, but some identified shortcomings in both content and form need to be addressed based on the specific recommendations below:
Shape suggestions
1. The bibliographic style used is not the one adopted by MDPI (bibliographic clues must be inserted in the text between square brackets [x] and never after a full stop within a sentence).
2. Sections and subsections should be numbered as in the example provided as a journal template.
3. Figure 1 should be modified at the 'reporting centers' term level to remove red underlining due to language settings.
4. It is advisable to edit the tables so that they are more readable and to avoid white writing of the text.
5. It is advisable to increase the readability and clarity of all figures because there is a lot of information in a small space and there are areas where readability is low.
6. Pages 8, 10, 12, 14 are more than half empty, so it is advisable to better organize the text and figures one after the other because a lot of space is lost.
7. Bibliographic sources number 10, 11, 14, 29, 36 are incomplete, missing data related to volume/issue/pages etc.
Content suggestions
1. The concluding part of the abstract should be improved in terms of results and future research directions to which this research can refer.
2. It is recommended to use more relevant keywords to help increase the visibility and importance of the paper (e.g., omicron, multicentre study, etc.).
3. The aim of the paper should be improved from the perspective of describing the contribution to the field under analysis and the elements of scientific novelty presented because the authors only presented in the last paragraph of the introduction what they did in the study.
4. The general management within COVID-19 should be briefly mentioned in order to provide the current picture. It would be advisable to briefly discuss the pathophysiological mechanism in order to understand from where the hematologic malignancies may arise, to present the most relevant comorbidities and associated pathologies, some proteins with diagnostic and prognostic roles, and last but not least, the treatment. I suggest checking and referring to the following updated sources: PMID: 36406478; PMID: 35131656; PMID: 32754599; PMID: 34863742.
5. It is advisable that the conclusions section be organized separately in this particular case to better capture the findings of this multicenter study and to better delineate the last part of the discussion addressing limitations and strengths.
Author Response
I attach a Word with reviewer's response

Round 2
Reviewer 1 Report
Comments and Suggestions for Authors
Authors are commended for their revisions. No further comments.
Reviewer 2 Report
Comments and Suggestions for Authors
The authors have significantly improved the manuscript based on the suggestions received.